# *CoT Referring*: Improving Referring Expression Tasks with Grounded Reasoning

## Abstract

Referring Expression Comprehension and Segmentation are critical tasks for assessing the integration of language understanding and image comprehension, serving as benchmarks for Multimodal Large Language Models (MLLMs) capabilities. To address these challenges, we propose a new strategy, ***CoT Referring***, which enhances model reasoning across modalities through a structured, chain-of-thought training data structure. Our approach systematically parses textual structures to a sequential referring step, where in each step it identifies relationships and ensures consistent reference alignment, thereby improving accuracy in complex query scenarios. We restructure the training data to enforce a new output form, providing new annotations for existing datasets and compiling an evaluation benchmark from existing resources. This benchmark is designed explicitly for complex referring cases. We also integrate detection and segmentation capabilities into a unified MLLM framework, training it with a novel adaptive weighted loss to optimize performance. Experimental results on our curated benchmark and the RefCOCO/+/g demonstrate the effectiveness of our approach, with a notable increase of 2.5%+ over baseline models.

## 1 Introduction

The advent of Multimodal Large Language Models (MLLMs) has revolutionized vision-and-language research, particularly in Referring Expression (RE) Tasks, such as Referring Expression Comprehension (REC) (Mao et al., 2016) and Referring Expression Segmentation (RES) (also known as Referring *Image* Segmentation) (Hu et al., 2016). This task is crucial as it demands a model's ability to integrate visual and textual information seamlessly. Recent advancements, such as (Li et al., 2024b; Rasheed et al., 2024; Ma et al., 2024; Zhang et al., 2024; Ren et al., 2024; Peng et al., 2023; Ma et al., 2023; Wang, 2022), have empowered Large Language Models (LLMs) to perform localization of specific objects in an image based on language queries.

While contemporary models excel at grounding simple object descriptions, their performance drops significantly when faced with complex referring expressions, as illustrated in Figure 2. As a result, we hypothesize this failure stems not from a fundamental inability to ground objects, but from a deficit in compositional reasoning. For a query like "the boy playing with a dog near the car," a correct answer requires a sequential reasoning process: one must first locate the "car," then identify the "dog" that is "near" it, and only then find the "boy" who is "playing with" that specific dog. Existing models, however, attempt to solve this in a single, holistic step, treating the query as a 'bag of objects' rather than a structured chain of dependencies. This reveals a fundamental mismatch: the task demands sequential logic, while the models provide a single-shot answer. This motivates our central research question: can we extend the model's response to explicitly model this sequential reasoning in referring expression tasks?

To address this challenge, we introduce the novel reasoning mechanism – ***CoT Referring*** (CoTR) – inspired by the **Chain-of-Thought (CoT) Prompting** used in LLMs (Wei et al., 2022; Kojima et al., 2022). This strategy enables models to parse complex input queries, understand the intended order of references, and reliably ground objects in the image. Compared to existing referring data where only the final mask is provided, CoTR data is more structured, as a chain-of-thought (CoT) reasoning process. To build it, we need to identify each relevant noun, termed *text anchors*, and order them to serve as an anchor point for subsequent references. Moreover, we notice insufficiency in existing

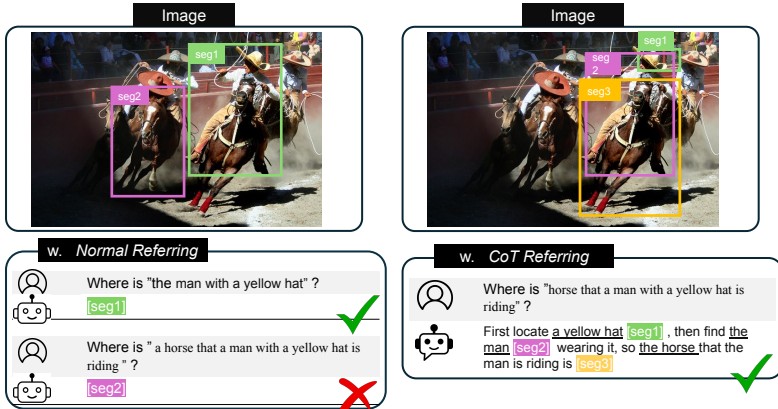

Figure 1: An example showcasing the answer of **RefLM** with and without ***CoT referring***. The underlined words in the answer are the anchors, and the highlighted words correspond to masks of the same color (for clarity, we visualize only the box).

data to both train and evaluate the performance on complex referring expressions. RefCOCO/+/g (Yu et al., 2016; Kazemzadeh et al., 2014b; Mao et al., 2016) only have an average text length of around 5.5 words, from which CoT reasoning might have little benefit on most of the cases. To address this, we propose a new evaluation benchmark specifically for *composite* referring expressions, i.e. expressions containing 3+ cross-related objects to find target. To bridge this gap, we utilize an LLM-based data pipeline (see Fig. 3) to filter and gather clean and complex referring expressions from RefCOCO/+/g datasets. The final training data and benchmark are about 20k and 3.7k referring expressions, respectively.

It is important to note that unlike *Phrase Grounding* (Karpathy et al., 2014) tasks that primarily focus on directly grounding relevant nouns in a given text (oftentimes a caption), CoT Referring organize the answer by grounding the anchors in a specified order that lead to the target object. For example, for the query "the boy playing with a dog near the car,", CoTR's order is "car, dog, boy" while phrase grounding outputs masks in order of "boy, dog, car".

To the best of our knowledge, this work is the first to explore how Referring Expression models reason through complex referring queries and how to inject a reasoning condition to improve localization on such challenging phrases. Furthermore, we also provide a new benchmark dataset to robustly evaluate the model's reasoning capabilities. We demonstrate that this approach is effective in improving target localization in complex referring expressions. To summarize, this paper makes the following key contributions:

1. Recognizing that current MLLMs often fail on complex referring queries, we propose a new reasoning strategy for Referring Expression tasks – ***CoT Referring*** – to improve localization on such challenging expressions.

2. We develop an innovative label pipeline for constructing CoT Referring data and propose an evaluation benchmark specifically for composite referring expressions, sourced from RefCOCO(+/g) (Kazemzadeh et al., 2014a; Mao et al., 2016; Yu et al., 2016).

3. We present ***RefLM***, an MLLM that demonstrates the effectiveness of our strategy on our composite referring benchmark and the RefCOCO(+/g) benchmark.

## 2 RELATED WORK

### 2.1 PHRASE GROUNDING

Phrase grounding, also known as *visual grounding*, aims to "ground" (i.e., localize) every relevant object in a text query in an image (Karpathy & Fei-Fei, 2015; Rohrbach et al., 2016; Plummer et al., 2016). Early works relied on merging pre-trained box proposal networks or object detectors to ground objects (Chen et al., 2020a; Li et al., 2020; Kamath et al., 2021). Some methods focus on

weakly- and self-supervised techniques to reduce dependence on expensive region annotations (Arbelle et al., 2021; Shaharabany et al., 2022; He et al., 2023), while others introduce contrastive learning approaches that improve phrase–object alignment and phrase grounding capabilities (Gupta et al., 2020; Li et al., 2021; Radford et al., 2021). Phrase grounding datasets are mainly derived from caption corpora that match nouns to regions, whereas referring-expression datasets provide descriptions of a single target object.

## 2.2 REFERRING EXPRESSION TASKS

Referring Expression (RE) tasks comprise referring expression comprehension (REC) and referring expression segmentation (RES), which locate a target described by a natural-language expression. REC predicts bounding boxes, whereas RES predicts pixel-wise segmentation masks. The primary datasets supporting RE research include Ref-COCO, RefCOCO+, RefCOCOg, and RefCLEF, all of which consist of manually annotated referring expressions (Kazemzadeh et al., 2014b; Mao et al., 2016; Yu et al., 2016). These datasets provide essential benchmarks but typically contain expressions that map to a single target, along with its ground-truth box or segmentation mask. They do not stratify queries by reasoning complexity. Furthermore, evaluations are commonly reported as single-metric IoU averages over entire datasets, which obscures how performance varies across different levels of expression complexity. As a result, standard RE evaluations provide limited insight into model reasoning.

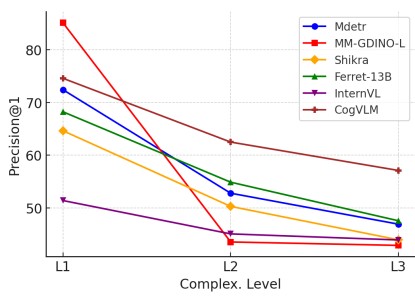

Figure 2: Model performance comparison across varying complexity levels of referring texts in FineCops-Ref positive subset. Complexity is defined by the maximum hop level from anchor noun to target. The performance clearly declines as complexity increases.

To address these limitations, additional datasets such as Ref-Reasoning, COPS-Ref, and FineCops-Ref (Yang et al., 2020; Chen et al., 2020b; Liu et al., 2024) have been introduced. These newer datasets capture the complexity of referring expressions by including compositional cases that require multi-step reasoning to locate the correct target. They stratify expressions by complexity and provide supervision for intermediate reasoning. However, they are primarily composed of template-generated language, leading to limited diversity, unnatural phrasing, and domain shifts.

## 2.3 MODELS FOR REFERRING EXPRESSION TASKS

Existing RE methods can be grouped into two families: specialist visual grounding models and MLLM-based models. Numerous specialist models are designed specifically for visual grounding tasks (Liu et al., 2023; Kamath et al., 2021; Chng et al., 2024; Zou et al., 2023; Wang et al., 2024; Chen et al., 2024b). These models typically employ dedicated architectures optimized for detection and segmentation.

MLLMs have seen rapid advances on vision–language tasks with detection and segmentation abilities. Recent developments (Ma et al., 2024; Ren et al., 2024; Rasheed et al., 2024; Chen et al., 2024c; Peng et al., 2023; Ma et al., 2023; Wang et al., 2023) focus on enhancing localization capabilities. Many approaches leverage instruction-following frameworks, often framing RE as instruction following with specialized heads for detection/segmentation. Some introduce special tokens (e.g., [SEG]) and require additional fine-tuning and alignment during training to handle box/mask generation. MLLMs (Ma et al., 2024; Ren et al., 2024; Rasheed et al., 2024; Chen et al., 2024c) also make use of pre-trained visual models and decoders to generate masks, such as ViT (Dosovitskiy et al., 2020) and SAM (Kirillov et al., 2023).

## 3 CURATION OF CoT REFERRING DATA

Existing referring expression (RE) datasets typically collapse multi-hop reasoning into a single flat answer, making models learn complex reasoning implicitly. Motivated by Chain-of-Thought (CoT)

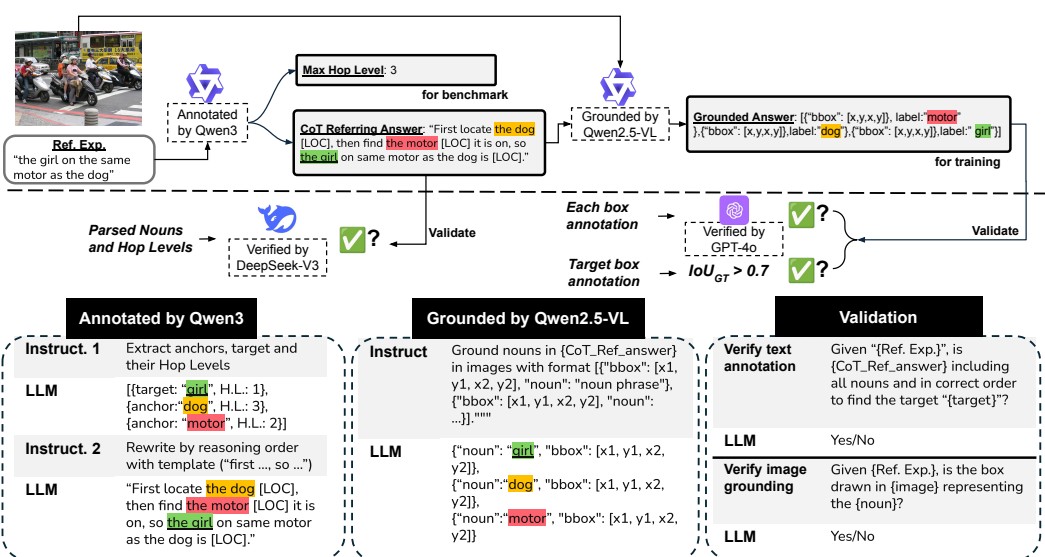

Figure 3: Data pipeline for generating CoT Referring training data and the Composite Referring Benchmark. Qwen3 first extracts anchors and the target and rewrites the CoT answer; parsed nouns and hop levels are validated by DeepSeek-V3. Qwen2.5-VL then grounds each noun with a bounding box. GPT-4o is used to verify each box; the target box must satisfy $IoU_{GT} > 0.7$. For the benchmark, the maximum hop level $L_{max}$ quantifies the compositionality of each referring annotation.

prompting, which shows that making intermediate steps explicit improves performance on reasoning tasks, we introduce **CoT Referring (CoTR)**, which reformulates referring expressions to explicitly guide models through the requisite reasoning steps.

## 3.1 DATA GENERATION PIPELINE

To generate CoTR data, we address three key research questions that cover both the textual decomposition into a chain-of-thought (CoT) and the visual grounding of each reasoning step: (Q1) How can a referring expression be decomposed into a canonical order of semantic anchors and a target? (Q2) How can each anchor be visually grounded at scale? (Q3) How can the difficulty of a composite referring expression be quantified for evaluation?

We address these questions with a four-stage data generation pipeline, depicted in Figure 3. This pipeline leverages a sequence of large language and vision-language models (Qwen3-235B-A22B (Qwen Team, 2025) → DeepSeek-V3.1 (DeepSeek-AI Team, 2024) → Qwen2.5-VL-72B (Bai et al., 2025) → GPT-4o (Hurst et al., 2024)). To quantify complexity of a referring express, we define the hop level of a anchor noun as its dependency distance to the target, and $L_{max}$, the maximum hop level among all anchors, as the complexity of the expression.

**Importance of Validation** Each stage includes validation. To demonstrate the effectiveness of these checks, we report stage-wise pass rates and an ablation removing validation in Appendix B.4. This rigorous process ensures high-quality annotations.

### 3.1.1 DATA FORMAT AND REPRESENTATION

**Notation and measures.** We use `[LOC]` as placeholders for spatial grounding, which are later mapped to bounding boxes. The hop level (H.L.) of a noun is the shortest dependency distance to the target in the parsed graph (the target has H.L.=1). We define complexity as $L_{max}$, the maximum hop level across nouns; larger $L_{max}$ indicates longer reasoning chains and more challenging compositional grounding (see Fig. 2). Following prior MLLMs (Rasheed et al., 2024; Peng et al., 2023; Ma et al., 2023; Zhang et al., 2024), we adopt an instruction-following format. A standard query like `"Where is the cat on the chair?"` might yield `"Cat on the chair is [LOC]"`, whereas

| Ref. Exp. | the white, glass, small and covered jar | a man hitting the ball | man with blue shirt and glove | a boy with a colorful hat is doing tricks on top of a skateboard | a man standing near the sheep touching a lying sheep with a rod | bicycle to the left behind the guy sitting in front of the motorcycle | a man in a white shirt with jeans taking a pic of a greyhound |
|---|---|---|---|---|---|---|---|
| Anchors (H.L.) | jar (1) | man (1), ball (2) | man (1), shirt (2), glove (2) | boy (1), hat (2), skateboard (2) | man (1), sheep (2), sheep (2), rod(3) | bicycle (1), guy (2), motorcycle (3) | man (1), shirt (2), jeans (2), pic (2), greyhound (3) |
| Max Hop Level | 1 | 2 | 2 | 2 | 3 | 3 | 3 |

Figure 4: Examples from our curated data. Each example shows the referring expression, its text anchors with hop levels (H.L.), and the corresponding noun groundings. Highlighted nouns correspond to mask of the same color in the image.

our CoTR format rewrites it to expose intermediate steps: `"First locate the chair [LOC], then the cat on top of it [LOC]."` This format promotes (1) a **canonical reasoning order** by decomposing the query into clear, interpretable steps and (2) **gradual visual grounding** by mapping each `[LOC]` placeholder to a bounding box, providing a strong supervisory signal for each step.

### 3.1.2 STAGE A: TEXTUAL DECOMPOSITION AND REWRITING

The first stage of our pipeline focuses on textual analysis and rewriting. The detailed prompts are in the Appendix B.1. **A.1 — Extraction.** Given a referring expression, Qwen3 (Qwen Team, 2025) parses the text to identify all nouns, assigns dependency hop levels (H.L.), and distinguishes between anchors and the final target. **A.2 — Reordering.** Based on the extracted dependency structure, Qwen3 (Qwen Team, 2025) then generates a CoT_Ref_answer. This rewritten expression places anchors before the target, sorted by hop level, and uses `[LOC]` placeholders for future grounding. We term this process *Rewriting by Reasoning Order*; it preserves linguistic fluency compared to template-based methods (Yang et al., 2020). **A.3 — Validation.** To ensure the quality of the rewritten expression, DeepSeek-V3 (DeepSeek-AI Team, 2024) validates the previous outputs. It checks two properties: coverage (i.e., all nouns from the original expression are present in the rewrite) and order consistency (i.e., the sequence of anchors respects the dependency hop levels). This yields a cleaned and logically sound CoT rewrite.

### 3.1.3 STAGE B: VISUAL GROUNDING

Having validated the text, we proceed to ground each noun and verify the results. **B.1 — Grounding.** Qwen2.5-VL-72B (Bai et al., 2025) is prompted to predict one bounding box for each `[LOC]` placeholder. To improve grounding quality, including cases with multiple instances of the same noun (e.g., "giraffe with head on another giraffe") or unclear/pronominal mentions (e.g., "theirs", "one"), we include the relevant span of each anchor in referring expression as context in the query prompt. This added context clarifies references and helps align textual steps with correct image regions. **B.2 — Verification.** We verify the generated boxes with two filters: (i) for each noun, we render its predicted box on the image and ask GPT-4o (Hurst et al., 2024) for a yes/no confirmation that the box correctly localizes the noun (GPT-4o is strong at visual verification but not precise localization, so best for verification), and (ii) for the final target object, we require $IoU_{GT} > 0.7$ against the dataset ground truth. A sample is accepted only if both conditions are satisfied.

### 3.2 GENERATED DATASETS

The pipeline yields two data products derived from the RefCOCO(+/g) dataset, with samples illustrated in Figure 4. From the `train` split, we generate CoTR data for Supervised Fine-Tuning (SFT); from the `val` and `test` splits, we construct the **Composite Referring Benchmark** to evaluate model performance on challenging expressions (large $L_{max}$). More detailes are in Appendix section B.4.

**Composite Referring Benchmark** We select expressions with a maximum hop level $L_{max} \geq 3$. After removing images with known annotation errors (Chen et al., 2024a), the final benchmark consists of approximately 1.8k images and 3.7k referring expressions (average 9.5 words and 3.7

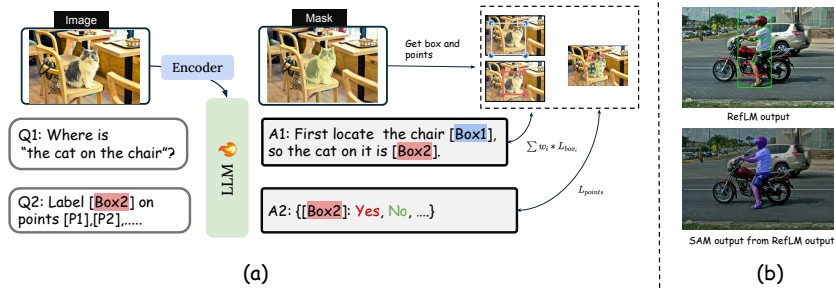

(a)                                                                          (b)

Figure 5: (a) **RefLM** architecture. The model consists of a vision encoder (including projector), and a language model (LLM). Visual prompts generated by the MLLM are fed into the Segment Anything Model (SAM) for mask generation. (b) An example of our model output, including the LLM output and the SAM output.

anchors per expression). All cases underwent manual verification to ensure label accuracy. This benchmark provides a rigorous test of a model's reasoning and grounding capabilities.

**CoTR SFT Data** This set includes all samples that pass validation, without filtering by complexity, yielding approximately 20k high-quality CoTR instances from RefCOCO(+/g) used for training.

## 4 METHODOLOGY

### 4.1 PROBLEM DEFINITION

Given an image $I$ and a referring expression $T$, we aim to learn parameters $\Phi$ of a model that predicts a bounding box $B_\Phi$ (REC) and a segmentation mask $\hat{M}_\Phi$ (RES), so as to maximize IoU with the ground truth box $B^\star$ and mask $M^\star$, respectively.

$$B_\Phi = g_\Phi(I, T), \quad \Phi_{\text{REC}}^\star = \arg\max_\Phi \text{IoU}(B_\Phi, B^\star) \tag{1}$$

$$\hat{M}_\Phi = f_\Phi(I, T), \quad \Phi_{\text{RES}}^\star = \arg\max_\Phi \text{IoU}(\hat{M}_\Phi, M^\star) \tag{2}$$

### 4.2 MODEL ARCHITECTURE

We propose **RefLM** (Fig. 5), which follows a typical MLLM architecture comprising a vision encoder, a projector, and an LLM. Unlike prior models (Rasheed et al., 2024; Zhang et al., 2024; Li et al., 2024b), RefLM removes the learned mask-decoder head. Instead, it predicts visual prompts (bounding boxes and points) that provide fine-grained spatial cues, and then uses the Segment Anything Model (SAM) to generate masks. This eliminates the need to learn an additional embedding-to-mask mapping (decoding [SEG] token) between the LLM and a mask decoder, because supervision can be applied directly at the token level.

### 4.3 BOX-POINT REPRESENTATION FOR GROUNDING

Conventional grounding Multimodal Large Language Models (MLLMs) introduce a special segmentation token that is projected and consumed by a mask decoder. As shown in Fig. 5, we instead have the Language Model (LLM) output a bounding box and Yes/No labels indicating whether sampled points lie on the target object. It encodes the box and point classifications as: "[x_min, y_min, x_max, y_max], Yes,Yes,No...". When a mask is required, a Segment Anything Model (SAM) decoder (Kirillov et al., 2023) is used as a post-processing step to convert these prompts into a mask. This removes the need to train an additional mask decoder. We empirically find this to be effective for grounding. How points are sampled within the box is a critical design choice. To improve robustness, we randomly sample points inside the target box during training and use evenly spaced points at inference. Sampling strongly influences mask quality since SAM is sensitive to the number of input points, which we analyze in Sec. 5. In our main evaluation, we form a $5 \times 5$ grid (25 points) within the box and use them for mask generation. Also, we normalize the bounding-box

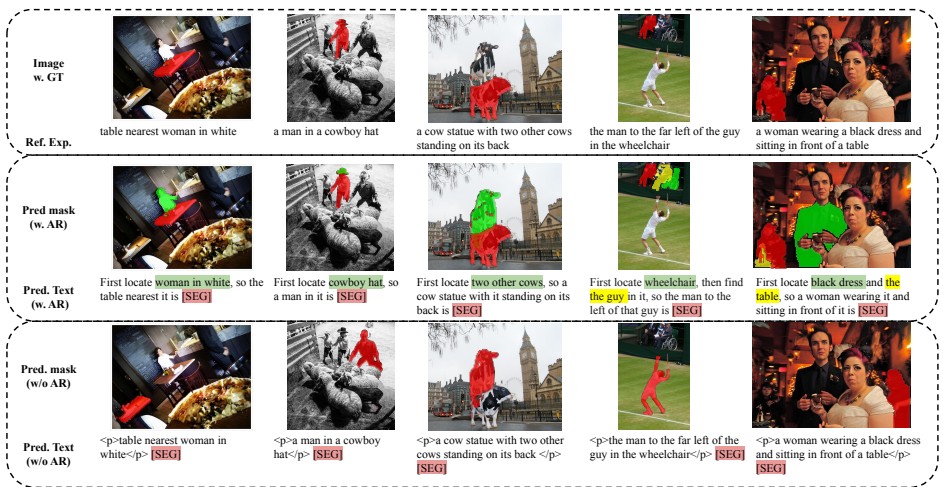

Figure 6: Comparison of RefLM with and without CoT Referring (CoTR) data. Colored highlights link anchors to their masks.

coordinates to $[0, 1000]$ so that, in the LLM tokenizer, each coordinate $(x, y)$ is encoded as a single token, yielding a total of $4 + 15 = 19$ tokens for a mask prompt.

### 4.4 ADAPTIVE WEIGHTED LOSS

In Chain-of-Thought Referring (CoTR), the model predicts a sequence of groundings: several intermediate anchors followed by the final target. Treating all boxes equally with a uniform cross-entropy (CE) loss can overemphasize anchor accuracy and underweight the true objective—the final target.

To bias learning toward the target while still supervising anchors, we apply an adaptive weighting to the box loss:

$$L_{\text{box}} = \sum_{i=0}^{n} w_i \text{CE}(\hat{B}_i, B_i), \quad w_i = \begin{cases} n+1, & i = 0 \text{ (final target)} \\ 1, & i > 0 \text{ (anchors)} \end{cases} \quad (3)$$

where $n$ is the number of intermediate anchors, and $B_0$ represents the final target box. This weighting ensures the final target box contributes more to the loss than any individual anchor box, thereby prioritizing the primary prediction. The point loss is defined as $L_{\text{points}} = \sum_{j=1}^{m} \text{CE}(\hat{P}_j, P_j)$, and the total loss is:

$$L = L_{\text{box}} + L_{\text{points}} + L_{\text{text}}. \quad (4)$$

where $\hat{P}_j$ and $P_j$ are the predicted and ground truth points, respectively, and $L_{\text{text}}$ is the cross-entropy loss associated with the text output of the LLM.

## 5 EXPERIMENTS

**Implementation Details.** For RefLM-8B, we initiate our weights from LLaVA-NeXT-8B (Li et al., 2024a). For RefLM-7B, we follow the setting of OMG-LLAVA-7B (Zhang et al., 2024), using Open-CLIP's ConvNeXT-L as the vision encoder (Liu et al., 2022), and InternLM2-7B as the LLM (Cai et al., 2024). This setup for RefLM-7B is designed to be directly comparable to other leading 7B models. For RefLM-8B, we only need to conduct instruction tuning, while for RefLM-7B, an extra alignment stage is needed, before instruction tuning, as mentioned in OMG-LLAVA (Zhang et al., 2024). SAM (Kirillov et al., 2023) is employed for mask prediction. For groundings of anchors, we use bounding boxes, while for targets we utilize box-point representation to learn mask representations. Training is conducted with an initial learning rate of 1e-4 and a global batch size of 128. The perception model's weights are frozen, and the LLM is fine-tuned using LoRA with a rank of 256 (Hu et al., 2021). All experiments are conducted on eight A100 GPUs with 40GB memory.

**Training Dataset.** The instruction tuning process of RefLM involves various grounding tasks, including both REC/RES and Grounding Caption Generation (GCG) (Rasheed et al., 2024). For GCG

Table 1: Results on the Composite Referring Benchmark (IoU@Box, gIoU@Mask). Best performance among the group is bolded.

| Method | Model Size | Max Hop Level | | | # of Anchors | | | Avg. |
|---|---|---|---|---|---|---|---|---|
| | | 3 | 4 | 5+ | 3 | 4 | 5+ | |
| **Specialist (Box)** | | | | | | | | |
| MDETR (Kamath et al., 2021) | <1B | 36.2 | 34.5 | 34.8 | 37.1 | 34.3 | 34.2 | 35.18 |
| MM-GDINO-T (Zhang, 2022) | <1B | 37.8 | 35.4 | 35.6 | 38.1 | 35.7 | 34.4 | 37.17 |
| MM-GDINO-T-FC (Liu et al., 2024) | <1B | 54.3 | 52.1 | 51.7 | 54.8 | 51.9 | 51.9 | 53.79 |
| **MLLM (Box)** | | | | | | | | |
| GPT4-V + SoM (Yang et al., 2023) | - | 61.5 | 58.7 | 58.2 | 62.3 | 59.1 | 57.8 | 60.60 |
| Ferret-13B (Li, 2023) | 13B | 63.4 | 60.2 | 60.1 | 64.2 | 60.8 | 59.5 | 61.37 |
| CogVLM-17B (Ding, 2023) | 17B | 68.3 | 65.7 | 65.5 | 68.8 | 65.2 | 65.2 | 66.85 |
| CogCom-17B (Huang, 2023) | 17B | 68.1 | 65.5 | 65.3 | 68.7 | 65.0 | 65.1 | 65.29 |
| Qwen2.5-VL-7B (Bai et al., 2025) | 7B | 68.1 | 67.2 | 65.6 | 69.2 | 65.8 | 64.5 | 66.97 |
| RefLM-8B | 8B | **68.1** | **67.4** | **66.8** | **69.1** | **66.3** | **66.0** | **67.28** |
| **Specialist (Mask)** | | | | | | | | |
| SEEM (Zou et al., 2023) | <1B | 50.4 | 48.1 | 48.3 | 51.2 | 48.3 | 47.7 | 49.84 |
| **MLLM (Mask)** | | | | | | | | |
| GLAMM-7B (Rasheed et al., 2024) | 7B | 61.2 | 60.3 | 60.6 | 61.9 | 59.5 | 59.2 | 60.95 |
| OMG-LLAVA-7B (Zhang et al., 2024) | 7B | 61.3 | 58.4 | 57.9 | 61.8 | 58.2 | 57.1 | 58.87 |
| READ-7B (Qian et al., 2025) | 7B | 62.0 | 58.9 | 56.5 | 62.3 | 58.7 | 56.8 | 59.20 |
| RefLM-8B w/o Reason. Order | 8B | 60.4 | 59.1 | 58.2 | 61.1 | 58.7 | 58.5 | 59.83 |
| RefLM-7B | 7B | 62.5 | 61.4 | 60.4 | 65.1 | 60.8 | 58.3 | 61.90 |
| RefLM-8B | 8B | **64.3** | **62.2** | **60.8** | **65.1** | **62.3** | **61.4** | **62.58** |

Table 2: Comparison on refCOCO, refCOCO+, and refCOCOg (cIoU@Mask). Gray models use significantly more data (∼500 times) or are larger.

| Method | Model Size | refCOCO | | | refCOCO+ | | | refCOCOg | |
|---|---|---|---|---|---|---|---|---|---|
| | | val | testA | testB | val | testA | testB | val(U) | test(U) |
| **Specialist** | | | | | | | | | |
| ReLA (Wang et al., 2024) | <1B | 73.8 | 76.5 | 70.2 | 66.0 | 71.0 | 57.7 | 65.0 | 66.0 |
| MagNet (Chng et al., 2024) | <1B | 77.4 | 80.9 | 74.7 | 69.4 | 76.1 | 61.4 | 69.0 | 71.7 |
| UniRES (Chen et al., 2024b) | <1B | 76.6 | 78.3 | 72.2 | 68.1 | 73.6 | 61.8 | 67.8 | 69.3 |
| **MLLM (Large)** | | | | | | | | | |
| GSVA-Llama2-13B (Xia et al., 2024) | 13B | 79.2 | 81.7 | 77.1 | 70.3 | 73.8 | 63.6 | 75.7 | 77.0 |
| **MLLM (Small)** | | | | | | | | | |
| GLaMM-7B (Rasheed et al., 2024) | 7B | 79.5 | 83.2 | 76.9 | 72.6 | 78.7 | 64.6 | 74.2 | 74.9 |
| LISA-7B (Li et al., 2024b) | 7B | 74.9 | 79.1 | 72.3 | 65.1 | 70.8 | 58.1 | 67.9 | 70.6 |
| PerceptionGPT-7B (Pi et al., 2024) | 7B | 75.1 | 78.6 | 71.7 | 68.5 | 73.9 | 61.3 | 70.3 | 71.7 |
| GSVA-7B (Xia et al., 2024) | 7B | 77.2 | 78.9 | 73.5 | 65.9 | 69.6 | 59.8 | 72.7 | 73.3 |
| OMG-LLAVA-7B (Zhang et al., 2024) | 7B | 78.0 | 80.3 | 74.1 | 68.7 | 73.0 | 61.6 | 71.1 | 71.9 |
| SESAME-7B (Wu et al., 2024) | 7B | 74.7 | - | - | 64.9 | - | - | 66.1 | - |
| READ-7B (Qian et al., 2025) | 7B | 78.1 | 80.2 | 73.2 | 68.4 | 73.7 | 60.4 | 70.1 | 71.4 |
| RefLM-7B | 7B | 78.2 | 81.0 | 74.5 | 68.1 | 73.7 | 62.5 | 73.0 | 73.1 |
| RefLM-8B | 8B | **79.0** | **82.4** | **75.9** | **73.1** | **74.2** | **63.5** | **75.9** | **77.2** |

tasks, we include the Flickr30K entity data (Young et al., 2014) and GranDf (Rasheed et al., 2024). For RE tasks, we include refCOCO(+/g) (Yu et al., 2016; Mao et al., 2016; Kazemzadeh et al., 2014b), totaling 74K data and the re-annotated CoTR data, about 20K after verification. We also include semantic segmentation datasets, ADE20K (Zhou et al., 2017), PACO-LVIS (Ramanathan et al., 2023) and Part-Imagenet (He et al., 2021), totaling 26K of image-mask annotations.

## 5.1 EXPERIMENTAL RESULTS

**Metrics.** We evaluate our model using two common metrics in Referring Expression Segmentation (RES). Mathematically, they are expressed as: $\text{cIoU} = \frac{\bigcup_i M^i_{\text{pred}} \cap \bigcup_i M^i_{\text{gt}}}{\bigcup_i M^i_{\text{pred}} \cup \bigcup_i M^i_{\text{gt}}}$ for cumulative IoU and $\text{gIoU} = \frac{1}{N}\sum_{i=1}^{N} \frac{|M^i_{\text{pred}} \cap M^i_{\text{gt}}|}{|M^i_{\text{pred}} \cup M^i_{\text{gt}}|}$ for generalized IoU. where $M^i_{\text{pred}}$ and $M^i_{\text{gt}}$ are the predicted and ground truth masks for the $i$-th sample, respectively. For Referring Expression Comprehension (REC), we use the IoU between the predicted and ground truth bounding boxes.

## 5.2 MAIN RESULTS.

**Composite Referring Benchmark.** Table 1 showcases performance on our Composite Referring Benchmark, evaluated by IoU@Box for detection models and gIoU@Mask for segmentation models. The benchmark is stratified by max hop level and the number of anchors to test reasoning

Table 3: Ablation studies for our proposed method. We analyze the impact of various components: (a) adding boxes for anchor nouns and reasoning order on CoTR data, (b) different weighted loss strategies, and (c) the number of points sampled for SAM predictions.

| (a) Component Ablations | | | | | | (b) Weighted Loss Strategies | | | | (c) No. of Sampled Points | | |
|---|---|---|---|---|---|---|---|---|---|---|---|---|
| Anchor Boxes | Reason. Order | Adaptive Loss | Point Sampling | cIoU | gIoU | Loss Strategy | Weight | cIoU | gIoU | Points | cIoU | gIoU |
| ✗ | ✗ | ✗ | ✗ | 70.01 | 71.78 | Normal | - | 73.83 | 75.47 | 5 | 75.9 | 75.7 |
| ✔ | ✗ | ✗ | ✗ | 71.21 | 73.52 | Fixed | 2 | 75.27 | 76.89 | 10 | 76.1 | 76.0 |
| ✔ | ✔ | ✗ | ✗ | 73.83 | 75.47 | Fixed | 3 | 75.34 | 76.92 | 15 | 77.2 | 78.4 |
| ✔ | ✔ | ✔ | ✗ | 75.90 | 77.20 | Fixed | 4 | 75.30 | 76.90 | 20 | 76.2 | 77.1 |
| ✔ | ✔ | ✔ | ✔ | 77.2 | 78.40 | Adaptive | - | 75.90 | 77.20 | 25 | 75.9 | 77.2 |

capabilities. The results show that our RefLM models outperform other methods. Notably, training RefLM-8B on CoTR data without an explicit reasoning order ("RefLM-8B w/o Reason. Order") performs worse, demonstrating improvements from CoTR is not simply from additional boxes supervision, but also the reasoning knowledge in it.

**Standard RES Benchmarks.** Table 2 presents our performance on standard RES benchmarks. Models highlighted in gray are trained with significantly more data or are much larger in size. Our models achieve state-of-the-art results among comparably-sized models, with bold values indicating the best performance in this group. Qualitative results are presented in Appendix E due to space limitations. We also test RefCOCOm benchmark, and the results are presented in the Appendix D.

## 5.3 ABLATION STUDY.

Visualizations of RefLM with and without CoTR data are shown in Figure 6. These cases demonstrate that training on CoTR data helps to improve the grounding capability of target objects by also grounding the anchor objects described in the input query. Table 3 shows our ablations of each component on RefCOCOg-Val dataset.

**Component Ablation.** Table 3 (a) presents a cumulative ablation study on the key components of our method. We start with a baseline model where none of our proposed components are enabled. Subsequently, we progressively integrate each component: anchor boxes, reasoning order, adaptive loss, and number of sampled points in inference. The results show a consistent and significant performance improvement with each addition. Notably, incorporating the reasoning order yields a large gain in cIoU (+2.62), while the adaptive loss provides a substantial boost of +2.07 cIoU. The full model with all components achieves the best performance, demonstrating the effectiveness of each proposed element.

**Weighted Loss Strategies.** Table 3 (b) compares different weighted loss strategies. "Normal" is a baseline with uniform weights. "Fixed" applies a constant weight to the target grounding loss. "Adaptive" refers to our proposed method of dynamically calculating weights as described in Sec. 4.4. As shown, the adaptive strategy outperforms the others, highlighting its effectiveness.

**Number of Sampled Points.** Table 3 (c) evaluates the number of point prompts randomly sampled from a $5 \times 5$ grid during inference. Performance peaks at 15 points and does not increase monotonically with more points, which we attribute to SAM's properties (e.g. SAM is sensitive to the number of input points). Accordingly, we use 15 points in our main experiments.

## 6 CONCLUSION

In this work, we introduce Chain-of-Thought Referring (CoTR), a new data representation to improve the grounding of complex and composite referring expressions in Multimodal Large Language Models (MLLMs). CoTR structures these expressions by breaking them down into sequential, solvable sub-tasks to guide the model's reasoning process. To effectively train on CoTR data, we introduce a box-point mask representation and an adaptive weighted loss scheme. To evaluate our approach, we curated a comprehensive Composite Referring Benchmark with diverse and complex samples. Our experiments show that with CoTR, fine-tuned MLLMs significantly improve performance on these challenging composite referring cases. We hope this work and our new benchmark will encourage further research into the reasoning capabilities of MLLMs.

## ETHICS STATEMENT

We adhere to the ICLR Code of Ethics and explicitly acknowledge it during submission. Our study uses only publicly available datasets (RefCOCO/+/g, Flickr30K Entities, GranDf, ADE20K, PACO-LVIS, and Part-ImageNet) under their respective licenses; we collect no new human-subject data and release no personally identifiable information. Our curated CoTR SFT data and Composite Referring Benchmark consist of derived annotations over existing images; we will release annotations and scripts (not images), requiring users to obtain the original datasets. Large language models were used to assist in data curation; prompts, costs, and validation procedures are documented in the Appendix. Potential risks include misuse for surveillance or invasive tracking; we discourage such applications and will provide usage guidelines and licensing terms that prohibit biometric identification and other harmful uses. We are not aware of conflicts of interest beyond the affiliations listed.

## REPRODUCIBILITY STATEMENT

We also provide the information needed to replicate our results. The data curation pipeline, definitions of the CoT Referring SFT data and the Composite Referring Benchmark, and validation criteria are detailed in Section 3 and Section 3.2, with dataset sizes in Section 3.2. Model objectives and losses are specified in Section 4.4; training settings, datasets, metrics, and compute details are given in Section 5 (Implementation Details). The Appendix includes prompt templates, cost analyses, validation ablations, and benchmark quality assessments in Appendix B.1, Appendix B.4, and Appendix C, along with additional qualitative results and failure cases in Appendix E. Upon publication, we will release all the code and data.

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

## A  USE OF LLM IN WRITING

We used large language models to polish the text in this paper, improving readability and reducing grammatical errors.

## B  DETAILS FOR DATA PIPELINE

This section provides supplementary details for the data generation pipeline described in Section 3. We include the specific prompts for anchor extraction and hop-level parsing (Stage A.1), rewriting by reasoning order (Stage A.2), and visual grounding (Stage B.1). Furthermore, we present an ablation study to quantify the impact of our textual and visual validation stages, along with quality assessment and detailed statistics of our Composite Referring Benchmark. These materials are provided to ensure the reliability and reproducibility of our work.

Table 4: Size of curated data and benchmark.

| Data | Size |
|---|---|
| CoTR training data | 20K samples |
| Composite Referring Benchmark | 3.7K samples |

### B.1  PROMPT SPECIFICATIONS FOR CoTR DATA PIPELINE

In our data generation process, we utilize prompts aligned with the pipeline stages in Section 3. The first prompt, $P_a$, supports **Stage A.1 (Anchor Extraction and Hop-Level Parsing)** as detailed in Section 3.1.2; its structure is shown in Listing 9. The second prompt, $P_r$, supports **Stage A.2 (Rewrite by Reasoning Order)** in Section 3.1.2, with details in Listing 10. The third prompt, $P_g$, supports **Stage B.1 (Visual Grounding)** in Section 3.1.3, shown in Listing 11.

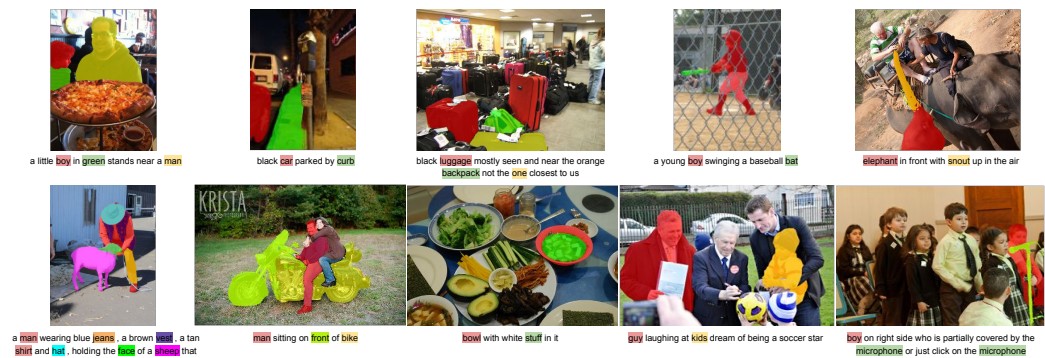

Figure 7: Examples of our curated data. For each example, we show the input image and the referring expression. The curated data includes the identified anchors and their dependency levels. The target object is highlighted, and the anchors are color-coded to match the corresponding nouns in the text. This visualization demonstrates how our pipeline parses complex expressions into a structured format suitable for training.

### B.2  MORE EXAMPLES FOR DATA CURATION

Figure 7 provides a gallery of examples for our curated data.

### B.3  COST OF USING LLMS IN DATA CURATION

In our data curation pipeline, we employed both open-source LLMs (Qwen3-235B-A22B-Instruct, Qwen2.5-VL-72B, and DeepSeek-V3.1) and closed-source LLMs (GPT-4o). We accessed open-source models through the OpenRouter API and closed-source models via the Microsoft Azure API.

**Training Data Costs.** For Stage A, we processed 31k samples and obtained 29k verified samples after validation. Each query required approximately 1,930 tokens (using Qwen3-235B-A22B-Instruct at $0.1 per 1M tokens), while verification consumed around 120 tokens per sample (using DeepSeek-V3.1 at $0.25 per 1M tokens). The total cost for Stage A was $6.66 (query: $5.98; verification: $0.93). For Stage B, we processed 29k samples and obtained 20k verified samples. Each query required approximately 690 tokens (using Qwen2.5-VL-72B at $0.07 per 1M tokens), while verification consumed around 540 tokens per sample (using GPT-4o at $2.5 per 1M tokens). The total cost for Stage B was $40.7 (query: $1.40; verification: $39.3). The complete training data curation pipeline cost approximately $47.36.

**Evaluation Data Costs.** For evaluation data, we only performed Stage A processing. We processed 4.5k samples and obtained 3.7k verified samples, with a total cost of $0.95.

### B.4 STUDY ON DATA VALIDATION STAGES

**Stage-wise Pass Rates.** We report stage-wise pass rates for each validation step in curating training data. Textual validation achieves an 89.3% pass rate. Visual validation attains a 70.3% overall pass rate, comprising an 89.4% pass rate for the GT IoU filter and 67.6% for GPT-4o verification.

**Ablation Study.** To quantify the impact of the textual and visual validation stages, we conduct an ablation study. We train our base model on four versions of the CoTR SFT dataset (20K samples each): (1) the final dataset produced by the full pipeline; (2) a version without the DeepSeek-V3.1 textual validation (Stage A.3 in Section 3.1.2); (3) a version without the GPT-4o visual verification and the IoU filter (Stage B.2 in Section 3.1.3); and (4) a version with no validation. As shown in Table 5, each validation step yields a substantial performance gain on RefCOCOg-Val, with the full pipeline performing best. Removing visual validation produces the largest degradation, underscoring the importance of accurate grounding of both anchor and target objects in the training data.

Table 5: Ablation study on the impact of data validation stages. Performance is measured by gIoU on the RefCOCOg-Val dataset.

| Text Validation | Visual Validation | cIoU |
| --- | --- | --- |
| - | - | 69.1 |
| ✓ | - | 70.5 |
| - | ✓ | 72.8 |
| ✓ | ✓ | **75.9** |

## C ASSESSMENT OF THE COMPOSITE BENCHMARK

This section provides supplementary details for the quality assessment and statistics of the Composite Referring Benchmark described in Section 3.2.

### C.1 QUALITY ASSESSMENT

We propose an automatic audit using Gemini 2.5 Pro and GPT-4o as independent judges of textual label quality. Judges verify: (i) coverage and correctness of anchors and the target, and (ii) correctness of hop-level assignments.

**Protocol.** We evaluate a stratified random sample (e.g., 1,000 expressions; balanced across $L_{max}=3/4/5+$ and RefCOCO/+/g) to control budget and variance. Each judge receives the original expression and the extracted anchors with hop levels, and returns yes/no decisions for:

- **Anchor coverage**: all and only the nouns in the expression are present and correctly typed in the rewrite.
- **Hop-level correctness**: assigned hop levels for each anchor and the target are correct.

We compute per-judge rates, inter-judge agreement, and Cohen's $\kappa$. A sample is *accepted* if both judges return yes on all checks.

The high yes rates ($\geq$98%), very high agreement ($\geq$99%), and $\kappa$ in the 0.83–0.94 range in Table 6 indicate that our Composite Referring Benchmark has high-quality labels on anchors and hop levels.

Table 6: Automatic audit of Composite Benchmark textual labels by Gemini 2.5 Pro and GPT-4o on a stratified sample. Values are percentages; Agreement is percent matching decisions; $\kappa$ is Cohen's kappa.

| Metric | Gemini-2.5-Pro | GPT-4o | Agreement | $\kappa$ |
|---|---|---|---|---|
| Anchor coverage (yes rate) | 99.2 | 99.0 | 99.7 | 0.83 |
| Hop-level correctness (yes rate) | 98.5 | 98.7 | 99.8 | 0.93 |
| Overall accept | 98.2 | 98.4 | 99.8 | 0.94 |

## C.2 STATISTICS

To focus on compositional reasoning, we prioritize longer expressions and require the maximum hop level $L_{\max} \geq 3$. The resulting subset emphasizes harder multi-hop cases; summary statistics appear in Table 7.

Table 7: Statistics comparison between val splits of RefCOCO(+/g) datasets and our Composite Benchmark. Our Composite Benchmark only includes expressions with max hops $\geq 3$.

| Metric | RefCOCO | RefCOCO+ | RefCOCOg | Ours |
|---|---|---|---|---|
| Avg words/sentence | 3.58 | 3.69 | 8.53 | **9.50** |
| Avg hops/sentence | 1.28 | 1.41 | 1.71 | **3.10** |
| Avg max hop level | 1.32 | 1.56 | 2.10 | **3.71** |
| Max Hop Level 3 (%) | 2.54 | 5.53 | 20.89 | 60.23 |
| Max Hop Level 4 (%) | 1.05 | 1.41 | 2.46 | 23.97 |
| Max Hop Level 5+ (%) | 0.81 | 1.01 | 1.11 | 16.80 |

## D REFCOCOM RESULTS

As shown in Table 8, our models also achieve superior performance on the RefCOCOm benchmark for part-level segmentation. The evaluation includes two settings: "Part" for part-only segmentation, and "Obj & Part" for segmenting both the object and its parts.

Table 8: Comparison on the RefCOCOm part-level segmentation benchmark (gIoU).

| Methods | val | | testA | | testB | |
|---|---|---|---|---|---|---|
| | Part | Obj & Part | Part | Obj & Part | Part | Obj & Part |
| *Specialists* | | | | | | |
| SeqTR (Zhu et al., 2022) | 13.9 | 28.2 | 12.1 | 22.8 | 18.1 | 34.7 |
| CRIS (Wang et al., 2022) | 10.6 | 25.4 | 10.1 | 21.2 | 12.9 | 30.0 |
| LAVT (Yang et al., 2022) | 15.3 | 29.9 | 13.2 | 24.4 | 18.7 | 35.5 |
| *Generalists* | | | | | | |
| X-Decoder (Zhang et al., 2023) | 16.2 | 29.5 | 13.6 | 23.6 | 20.3 | 33.8 |
| SEEM (Zou et al., 2023) | 16.1 | 29.4 | 13.6 | 23.4 | 20.4 | 33.9 |
| UniRES (Chen et al., 2024b) | 19.6 | 34.3 | 16.4 | 27.8 | 25.2 | 41.7 |
| RefLM-7B | 20.1 | 34.7 | 16.8 | 28.3 | 25.7 | 42.1 |
| RefLM-8B | **21.8** | **36.2** | **18.1** | **29.3** | **27.8** | **43.2** |

## E QUALITATIVE RESULTS FOR REFCOCO(+/G)

Figure 8 shows a qualitative comparison with GLAMM and OMG-Seg, demonstrating how RefLM excels at grounding both anchor and target objects in complex referring expressions.

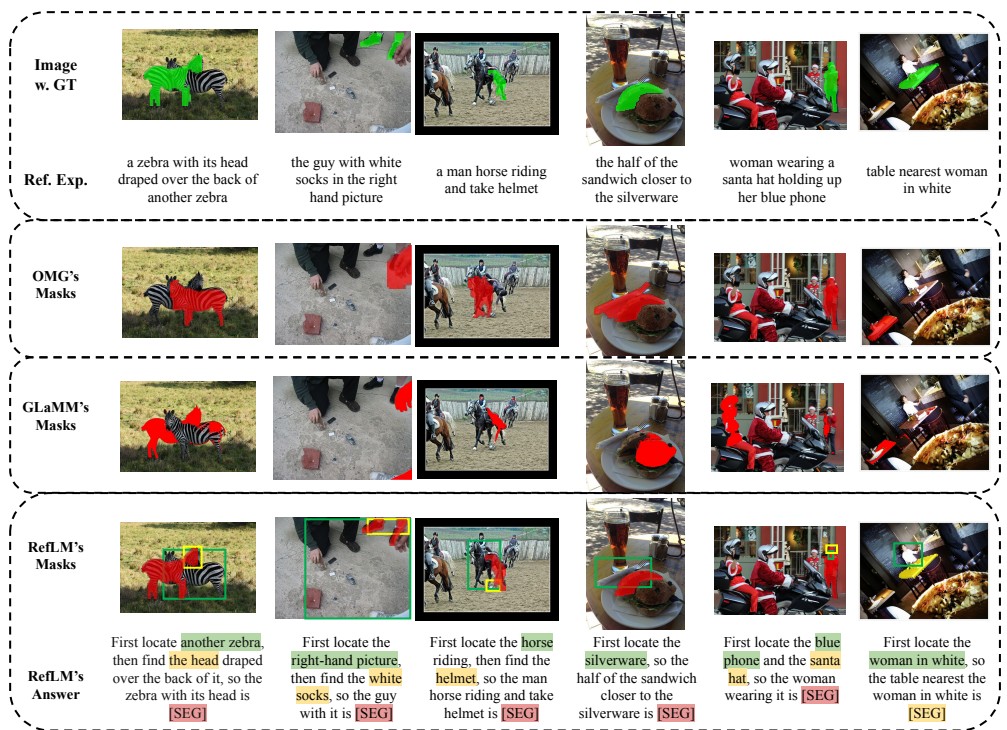

Figure 8: Qualitative comparison with GLAMM and OMG-Seg. RefLM excels at grounding both anchor and target objects in complex referring expressions.

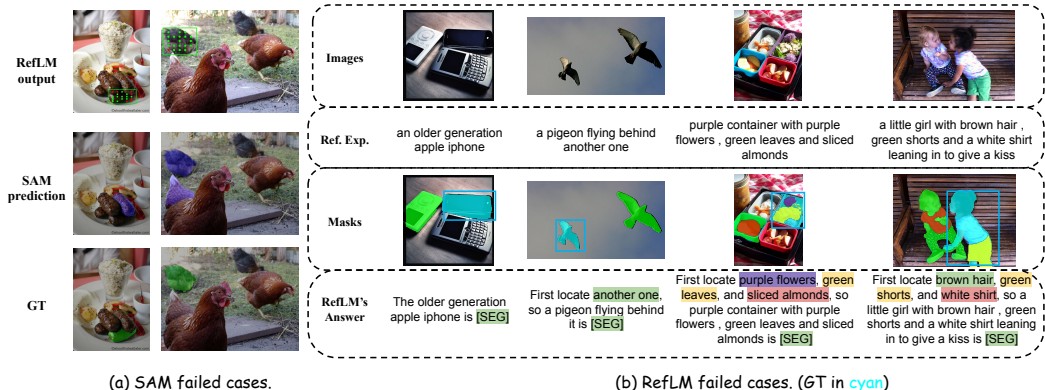

(a) SAM failed cases.

(b) RefLM failed cases. (GT in cyan)

Figure 9: Failed cases from RefLM. The figure illustrates two major types of challenging cases: (1) anchor grounding errors, where the model incorrectly identifies objects similar to the anchors, and (2) single noun case failures, where the model fails to detect the target object when only one noun is present. The predicted masks are shown in green boxes and masks, while ground truth annotations are displayed in cyan boxes and masks. Additional colored masks indicate anchor objects mentioned in the referring text, and identical colors are used when multiple nouns refer to the same object.

# F  FAILED CASES VISUALIZATION

Figure 9 illustrates examples of failed cases from RefLM. The first type of error occurs when there is only a single noun in the text, and the model fails to detect the target object. The second type of error is an anchor grounding error. In this scenario, although the model correctly segments most anchors mentioned in the text, it may mistakenly identify and select objects similar to a specific anchor, failing to identify the target object. This is demonstrated in columns three and four of Figure 9.

Table 9: Stage A.1 — Anchor Extraction and Hop-Level Parsing Prompt

---

**System Prompt**

---

You are a precise linguistic parser for referring expressions. Return only valid JSON. Assign noun-level dependency hops with: level 0 = target head noun(s); level k = shortest noun-to-noun hop distance via relations (with/behind/between/near/inside/on/next to/under/in front of/of/possessive/apposition/relative-clause/group/count). IMPORTANT: Directional/spatial words (left, right, top, bottom, center, corner, side, front, back, etc.) are modifiers, NOT anchor nouns. Only count concrete object nouns as anchors. Adjectives are not nouns. Spans are token index ranges [start, end) over the provided tokens.

**User Prompt**

---

Provide a JSON object with keys: sentence, tokens, nouns, target_indices.
- sentence: original string.
- tokens: use the tokens provided below (do not re-tokenize).
- nouns: array of objects {text, start, end, level}.
    start/end are token indices (start inclusive, end exclusive).
- target_indices: indices into nouns array for all level-0 nouns.

sentence: {sentence}
tokens: {json_tokens}

Identify all OBJECT nouns only (exclude spatial words like left/right/top/bottom). Mark target head noun(s) as level 0, and assign hop levels to required dependent object nouns.

**Example Output:**
```
{
    "sentence": "boy on girl with red skirt",
    "tokens": ["boy", "on", "girl", "with", "red", "skirt"],
    "nouns": [
        {"text": "boy", "start": 0, "end": 1, "level": 0},
        {"text": "girl", "start": 2, "end": 3, "level": 1},
        {"text": "skirt", "start": 5, "end": 6, "level": 2}
    ],
    "target_indices": [0]
}
```
**Key Rules:**
- Only concrete object nouns (exclude spatial/directional modifiers)
- Level 0: target head noun(s)
- Level k: shortest dependency hop distance from target
- Use provided token indices exactly (start inclusive, end exclusive)

---

Table 10: Stage A.2 — Rewrite by Reasoning Order Prompt

| System Prompt |
| --- |
| You are an assistant that analyzes referring expressions to identify the reference depth of objects and generates clear segmentation answers. Return only valid JSON following the exact schema. |

**User Prompt**

**RULES:**
1. Reference Depth Assignment:
   - The target (main) object is reference_depth = 0
   - Supporting objects get increasing depths based on steps away from main object

2. Segmentation Tag Rules:
   - Use `<seg n1>`, `<seg n2>`, etc. for supporting objects in the explanation
   - Always use `<seg main>` for the main target object

3. Answer Structure:
   - Start with "The target is [target noun]"
   - Explain steps from furthest to closest to target. You could use "first ...then ... so the [target noun] is ...".
   Note: if the referring text contains only one object, the answer is "[target phrase] is [SEG]".

**EXAMPLE 1:**
Expression: "Between the lamp and the vase on the table is the book"
Objects:
- lamp (n1), depth: 1
- vase (n2), depth: 1
- table (n3), depth: 2
- book (main), depth: 0

**Output 1:**
```
{
    "answer": "Target is the book. First locate <seg n3>the table</seg>, then find
    <seg n1>the lamp</seg> and <seg n2>the vase</seg> on it, and therefore, the book
    positioned between them is <seg main>the book</seg>."
}
```
Expression: "{text}"
Objects and their positions:
{objects}

Please respond with JSON format only.

Table 11: Stage B.1 — Visual Grounding Prompt (Qwen2.5-VL)

**System Prompt**

You are a precise visual grounding model that localizes object nouns in images. Return only valid JSON following the exact schema. All bounding boxes are in xyxy format [x1, y1, x2, y2] using absolute pixel coordinates. Clip all coordinates to image bounds. Include confidence scores in [0, 1] range.

**User Prompt**

Task: Ground all object nouns in the image with bounding boxes in pixel xyxy format.

Return only valid JSON with:
```
{
  "noun_bboxes": [
    { "noun_index": int, "text": string,
    "bbox_xyxy": [x1, y1, x2, y2],
    "confidence": number }
  ]
}
```
**Constraints:**
- Coordinates are absolute pixels within [0, {image_width}] × [0, {image_height}].
- One entry per noun index (length must equal {len(parsed_nouns)}).
- Clip boxes to image bounds.
- Include confidence score in [0, 1] for each detection.

**Context Enhancement:**
For improved grounding quality, each noun is provided with surrounding context from the CoT reasoning to disambiguate multiple instances and unclear references.

**Input Format:**
Image size: width × height = {image_width} × {image_height}
Full sentence: "{sentence}"

Parsed nouns with context:
- Index {i}: **{noun_text}** (level {level})
    Context: {surrounding_tokens_window}

**Output:**
Produce JSON for noun_bboxes with xyxy boxes for indices 0..{len(parsed_nouns)-1} and confidences in [0,1].

