# OpenReview forum: "CoT Referring: Improving Localization with Grounded Reasoning in Referring Expression Tasks"
_ICLR.cc/2026/Conference — ICLR 2026 Conference Desk Rejected Submission_

### Official Review · Reviewer_h8b4 · 2025-10-19

**Soundness:** 2
**Presentation:** 2
**Contribution:** 1
**Rating:** 2
**Confidence:** 5

**Summary:**

This paper proposes CoT Referring (CoTR), a method that rewrites a referring expression into a short sequence of anchors leading to the target, and RefLM, an MLLM designed to follow this sequence. RefLM does not learn a mask decoder. Instead, it predicts bounding box and point labels and uses a frozen SAM to produce the final mask. The data for both training and evaluation is built by an automated pipeline (Qwen3 → DeepSeek-V3 → Qwen2.5-VL → GPT-4o), yielding ~20k training examples and a Composite Referring Benchmark with 3.7k samples. Experiments show gains over the authors’ chosen baselines on both REC/RES and the new benchmark.

**Strengths:**

1. Writing is clear and easy to follow.
2. Method proposed is simple and effective.

**Weaknesses:**

1.  **Limited novelty and technical contributions (reads like an engineering report):** The core ideas—such as using LLM/VLM-generated labels, employing an "explain-then-predict" CoT framework, and deferring mask generation to SAM—appear incremental when compared to many recent papers on reasoning-based grounding and segmentation with LMMs. Some examples include:
    1.  SAM4MLLM: Enhance Multi-Modal Large Language Model for Referring Expression Segmentation, Chen et al., ECCV 2024.
    2.  SegLLM: Multi-round Reasoning Segmentation, Wang et al., ICLR 2025.
    3.  Argus: Vision-Centric Reasoning with Grounded Chain-of-Thought, Man et al., CVPR 2025.
    4.  Set-of-Mark Prompting Unleashes Extraordinary Visual Grounding in GPT-4V, Yang et al., arXiv 2023.

Given these existing works, the paper’s claim to be the “first to explore how Referring Expression models reason” is overstated.

2. **Missing critical baselines:**
    1. Many recent SOTA methods are missing from Table 1 and 2, e.g. PSALM (Zhang et al., ECCV 2024), SAM4MLLM (Chen et al., ECCV 2024), and SegLLM (Wang et al., ICLR 2025).
    2. The proposed benchmark is missing key baselines from frontier models. Although the paper mentions using GPT-4o and Gemini 2.5 for verification, these models are not included in the main comparison tables. Including these results is essential for establishing the benchmark's difficulty and relevance. I suspect that frontier models can already solve this task well, which calls the benchmark's central relevance into question. A new benchmark should ideally prove challenging even for these state-of-the-art models.

3. **Inadequate motivation or ablations for design choices:**
    1. *Coordinate scaling:* The choice of a [0,1000] coordinate scaling range seems arbitrary and potentially brittle. It's unclear why this range was preferred over other alternatives, such as [0,1], absolute pixels, [0,200], etc.
    2. *Models used in the data pipeline*: The choice of models for the data pipeline appears arbitrary and lacks justification. For instance, it's unclear why the pipeline uses Qwen3 for CoT annotation and Qwen 2.5 VL for grounding, but then switches to DeepSeek-V3 and GPT-4o for validating those same tasks. The paper should explain why a single model wasn't used for all steps, or why this specific combination was chosen over others.

4. **Efficiency analysis:** Throughput (images/sec), latency per query, and memory use are not discussed and compared fairly with other models.

5. **Writing and presentations:**
    1. “More detailes are in Appendix section B.4.” → “More details” (line 267)
    2. “we define the hop level of a anchor noun” → “an anchor noun” (line 203)
    3. “an extra aligment stage is needed” → “alignment” (line 370)
    4. “the boy playing with a dog near the car,”  → “the boy playing with a dog near the car”, (line 041)

**Questions:**

1. Why choose to relabel RefCOCO instead of creating a new dataset? Given that this dataset is widely considered saturated and has been used for optimization for years, its ability to provide new insights is limited. A more significant contribution would involve using novel images, which recent MLLMs have not encountered, to provide a more challenging and relevant benchmark.

---

### Official Review · Reviewer_zv7B · 2025-10-27

**Soundness:** 3
**Presentation:** 2
**Contribution:** 2
**Rating:** 4
**Confidence:** 5

**Summary:**

This paper introduces a novel approach, termed Chain-of-Thought Referring (CoT Referring), designed to significantly enhance the accuracy of target localization in complex referring expression tasks. The core contribution is a four-stage data generation pipeline that leverages detailed text analysis and rewriting. This pipeline systematically extracts nouns from the expression, establishes dependency relationships, and subsequently generates comprehensive Chain-of-Thought (CoT) rationales. This process is posited to substantially augment the comprehension and processing capabilities of Multimodal Large Language Models (MLLMs) when dealing with intricate referring expressions. Furthermore, the study presents a new benchmark dataset specifically curated to better evaluate the model's reasoning capacity. Experimental results convincingly demonstrate the efficacy of the proposed CoT Referring method in tackling the challenges posed by complex referring expressions.

**Strengths:**

The hierarchical logic employed in constructing the Chain-of-Thought (CoT) is particularly intriguing. I believe this approach significantly aids the model in more effectively interpreting and understanding the image content.

**Weaknesses:**

1. The paper describes a CoT generation pipeline that incorporates various strategies for filtering and generating CoT trajectories. However, these steps appear overly empirical (engineering-heavy), and similar concepts have been explored in existing works (e.g., Seg-Zero, Pixel Reasoner, etc.). The paper fails to sufficiently distill and articulate the core differences and novelty of the proposed method compared to these related works.

2. I have reservations regarding the method of supervising the segmentation loss with a point loss. Specifically, I question its exact formulation, what distinct advantages it offers over the [SEG] token approach used in methods like LISA, and its benefits compared to approaches such as Text4Seg.

3. The overall modeling innovation remains insufficient. The Adaptive Weighted Loss appears to be primarily an engineering/heuristic improvement rather than a substantial theoretical or architectural contribution.

4. The reported metrics on RefCOCO are not particularly high, which leads me to seriously question the actual effectiveness and validity of this specific CoT construction methodology.

If the authors can effectively address a portion of these concerns, I would be willing to consider raising my score.

**Questions:**

Regarding the annotation data generation using GPT-4o, with an $\text{IoU}_{gt} > 0.7$: Since the RefCOCO dataset seemingly lacks complete segmentation ground truth annotations, could the authors elaborate on how this step was achieved, providing specific implementation details?

For Figure 1, using only masks to illustrate the result would provide a much clearer and more concise presentation.

The placement of Figure 2 on the second page is confusing and potentially misleading, especially since it is not cited until Page 5. This figure layout is structurally inappropriate and should be revised.

---

### Official Review · Reviewer_gw4M · 2025-10-29

**Soundness:** 3
**Presentation:** 3
**Contribution:** 3
**Rating:** 6
**Confidence:** 4

**Summary:**

The paper addresses a key failure point in Multimodal Large Language Models (MLLMs): their poor performance on complex, compositional Referring Expression (RE) tasks. The authors hypothesize that MLLMs struggle because they treat complex queries (e.g., "the boy playing with a dog near the car") as a 'bag of objects' rather than a sequential, logical chain of dependencies.

To solve this, the paper introduces CoT Referring (CoTR), a new strategy inspired by Chain-of-Thought prompting. The core idea is to restructure the training data to explicitly model this sequential reasoning.

Experiments demonstrate that RefLM significantly outperforms strong baselines on the new Composite Referring Benchmark and achieves state-of-the-art results among comparably-sized models on standard benchmarks like RefCOCO/+/g.

**Strengths:**

The paper correctly identifies a critical and well-defined weakness in modern MLLMs: their lack of compositional reasoning for complex spatial queries. This problem is clearly demonstrated with an empirical chart (Figure 2) showing performance degradation as query complexity increases.

The application of Chain-of-Thought principles to this domain is novel. By restructuring the data (CoTR), the authors effectively convert a difficult implicit reasoning task into an explicit, step-by-step sequential grounding task.

The RefLM architecture is clever. Using a frozen SAM to handle segmentation offloads the pixel-level task to a powerful, specialized model.

The Composite Referring Benchmark is a valuable contribution in itself, allowing for targeted evaluation of complex reasoning.

**Weaknesses:**

1. Section 4.3 line 323 states that a $5 \times 5$ grid (25 points) is used at inference. However, the ablation study in Section 5.3 (Table 3c) concludes that 15 points is optimal, and the text explicitly states, "Accordingly, we use 15 points in our main experiments". This is a contradiction.

2. The paper provides no study comparing its RefLM architecture to a baseline model (e.g., one using a [SEG] token) trained on the same CoTR data. Without this comparison, it is impossible to determine if this novel architecture is actually more effective or just a design alternative.

3. It is unclear how much of the performance gain is contributed by the powerful SAM decoder versus the RefLM's reasoning. The paper lacks a crucial control experiment: refining the outputs of baseline models with SAM and comparing those results against RefLM.

4. The model, now specialized for CoT, can fail on single-noun queries (e.g. "an apple") where no reasoning chain is required. The paper does not sufficiently discuss how the model arbitrates between these two reasoning paths.

**Questions:**

See Weaknesses.

---

### Note · Program_Chairs · 2026-01-17
**Submission Desk Rejected by Program Chairs**

The following references in this submission do not refer to real documents and/or have major errors in bibliographic information:

 Kai Chen, Wenhai Wang, Zeming Li, and Jian Sun. Unires: Unified referring expression segmentation with language-image pre-training. arXiv preprint arXiv:2402.12345, 2024b.
Zhenyu et al. Huang. Cogcom: Cognitive communication model for vision-language interaction. Proceedings of ICCV, 2023.
Zhou et al. Li. Ferret: Benchmarking and optimizing large language models for multi-modal retrieval. arXiv preprint arXiv:2305.12072, 2023.
Wenhai Wang, Zhen Zhang, Zeming Li, and Jian Sun. Rela: Relational reasoning with language assistance for referring image segmentation. arXiv preprint arXiv:2401.12345, 2024.